# Role of Heat-Shock Proteins in the Determination of Postmortem Metabolism and Meat Quality Development of DFD Meat

**DOI:** 10.3390/foods13182965

**Published:** 2024-09-19

**Authors:** Muawuz Ijaz, Xin Li, Chengli Hou, Zubair Hussain, Dequan Zhang

**Affiliations:** 1Institute of Food Science and Technology, Chinese Academy of Agricultural Sciences/Key Laboratory of Agro-Products Quality & Safety in Harvest, Storage, Transportation, Management and Control, Ministry of Agriculture and Rural Affairs, Beijing 100193, China; muawuz.ijaz@uvas.edu.pk (M.I.); houchengli@163.com (C.H.); zubairaja530@gmail.com (Z.H.); dequan_zhang0118@126.com (D.Z.); 2Department of Animal Sciences, University of Veterinary and Animal Sciences, Jhang Campus, Jhang 35200, Pakistan

**Keywords:** beef, HSP, DFD, dark cutting, meat quality

## Abstract

This research explored the potential role of various heat-shock proteins (HSPs) in the determination of postmortem metabolism and the development of meat quality of normal, atypical DFD, and typical DFD beef. Beef *longissimus thoracis* muscle samples were classified into normal, atypical DFD, and typical DFD beef. The HSP27, HSP70, and HSP90 levels, meat quality parameters, and glycolytic metabolites were tested. The results showed that color coordinates (L*, a*, and b*), glycogen, and lactate contents were lower, whereas water-holding capacity was higher in the typical DFD beef than in the normal and atypical DFD beef (*p* < 0.05). The expression of HSP27 on day 1 was higher in atypical DFD beef. However, expressions of HSP70 on days 1 and 3 were higher in typical DFD, while the expression of HSP90 on day 1 was higher in atypical and typical DFD compared to the normal beef (*p* < 0.05). Interestingly, the expression of HSP27 was positively correlated with shear force readings. HSP70 and HSP90 presented a direct correlation with pH and water-holding capacity and an indirect correlation with a* and b*, glycogen and lactate contents (*p* < 0.05). The study concluded that the heat-shock proteins could influence the formation of DFD beef possibly by regulating the development of postmortem metabolism and meat quality traits.

## 1. Introduction

Heat-shock proteins (HSPs) are categorized under chaperone proteins, and their molecular weight ranges from 15 to 90 kDa [1]. Numerous investigations have demonstrated that HSPs play a protective function in cells and that their levels are obvious in response to detrimental circumstances such as high temperatures (hence the term “heat shock”), hypoxia, and dangerous oxidants [2,3,4]. HSPs play a crucial role in protein folding, unfolding, and the refolding of damaged proteins as molecular chaperones [2]. They are also known to be involved in the maintenance of cell survival by protecting the proteins from fatally aggregating substances in stressed conditions [5]. HSPs attach themselves to cell membranes and preserve their stability and integrity [6].

Medical science has given a lot of attention to the functions of HSPs; however, the connection between meat quality and HSPs was not clearly defined until recently. Numerous studies have consistently documented how various small heat-shock proteins affect the flavor, tenderness, water-holding capacity, and color of meat [4,7,8,9,10]. For example, the expression of HSPs in postmortem muscles is positively correlated with water-holding capacity. In pig muscles, decreased drip loss is known to be correlated with an increased abundance of the HSP70 [3]. Yu et al. [11] examined the impact of varying transportation durations on meat quality and the levels of HSP27, HSP70, HSP90, and alpha-B-crystalline. They discovered that the decrease in HSP expression was linked with increased drip losses. Pulford et al. [9] examined the distribution of small heat-shock proteins inside cells in postmortem beef with different meat pH levels and found that the ultimate meat quality is influenced by HSP 20 and HSP 27 in postmortem muscle. The current study focused on the role of HSP27, HSP70, and HSP90, and these specific proteins were selected based on findings from our previous experiments [12], which indicated their significant roles in the meat quality development of dark firm dry (DFD) beef.

DFD is directly associated with high pH values (pH > 5.8) and is a result of insufficient muscle glycogen at slaughter. Intramuscular glycogen has been shown to be below 66 μmol glucose g^−1^ in beef carcasses with muscle pH more than 6.0 [13]; on the other hand, bovine muscles with glycogen levels below 55 μmol glucose g^−1^ may likewise produce final pH values below 5.75 [14]. According to the research by Holdstock et al. [15] and Mahmood et al. [16], atypical DFD carcasses are beef carcasses that were graded dark at 24–48 h postmortem and had glucidic potentials that were comparable to normal beef or USDA Select carcasses. Furthermore, they revealed that slow pH decreases early postmortem, and poor glycolytic activity might be the cause of atypical DFD.

DFD is primarily a defect of meat color that can vary from dark purple to almost black, which is unacceptable to consumers. Different countries have different approaches to identifying DFD color [17]. For instance, Canada uses a plastic square color scheme, USDA utilizes the D^0^ and E^0^ ratings, and Australia uses color chips for color grading [17]. Furthermore, DFD meat has a great variation in tenderness [15,17,18]. Specifically, DFD meat that has a high ultimate pH (>6.2, typical DFD) is equal to or more tender than normal meat that has a pH of around 5.5. However, meat produced in the pH range of 5.8 to 6.2 (atypical DFD) is less tender than normal and typical DFD beef [15,16,17]. DFD is considered to arise from antemortem stress conditions in animals, leading to postmortem biochemical changes [16,19]. Although earlier studies have demonstrated the impact of HSPs in the determination of meat color, tenderness, flavor, and water-holding capacity, it is still unknown precisely how HSPs contribute to the development of DFD beef. Consequently, more research is necessary to determine the exact mechanism by which these HSPs affect the postmortem biochemical changes in the context of the formation of DFD beef.

Regarding sampling of the carcasses, the longissimus muscle was selected as it is the longest muscle in the animal body. According to Gajaweera et al. [20], *longissimus thoracis* (LT) is the most commonly utilized reference muscle in meat quality tests. Secondly, it is a representative cut for consumer-relevant quality studies since it is one of the most popular and commercially significant beef cuts [13,15]. The formation of DFD beef is mostly attributed to changes in stress, postmortem metabolism, and pre-slaughter handling, all of which have been shown to be especially susceptible to LT muscles in several investigations [15,16]. Therefore, we selected LT muscle for its susceptibility to DFD and its relevance in assessing overall meat quality.

The current study was designed to explore the association of HSP27, HSP70, and HSP90 in the determination of postmortem metabolism and development of other meat quality attributes of LT muscles of normal, atypical DFD, and typical DFD beef. We hypothesized that there is a significant link between the expression levels of HSP27, HSP70, and HSP90 and postmortem metabolic processes. Additionally, we believe that the development of distinct meat quality attributes is influenced by the differential expression of these proteins, especially in atypical DFD and typical DFD beef when compared to normal beef.

## 2. Materials and Methods

### 2.1. Sampling and Experimental Design

We conducted the current experiment using the samples from our previous study [21]. Briefly, sixty Simmental crossbred bull carcasses were selected from a neighboring slaughterhouse. The animals were 18–24 months of age with the same sex, breed, feeding system, batch, and pre-slaughter treatments. All the animals were slaughtered on the same day following the Halal slaughtering procedure, which involved exsanguination without electrical stunning. The carcasses were cooled to 4 °C. Within 24 h following the slaughtering, the pH of the carcasses was determined from the same side by a pH meter from three distinct, random places of the *longissimus thoracis* (LT) muscles. The pH of LT muscles was measured between the 11th and 12th ribs. As mentioned in our previous experiment [21], all sixty carcasses were ordered based on the average pH of triplicate measurement and eighteen carcasses were selected and categorized into three groups: normal (pH ≤ 5.70, *n* = 6), atypical DFD (5.70 ˂ pH ≤ 6.09, *n* = 6), and typical DFD (pH > 6.09, *n* = 6), based on the methods of [15,16]. The samples used in this study are meat that was purchased from the local slaughter house. There is no ethic issue in this study.

The LT muscles were removed from the left side of the chosen carcasses, wrapped in polyethylene bags, boxed with ice bags, and transported to the Institute of Food Science and Technology (IFST), Chinese Academy of Agricultural Sciences (CAAS), Beijing, within 3 h after collection. Upon arrival at IFST, the pH of the LT muscles was measured again to confirm the ultimate pH and their respective group. From the posterior end of LT muscles, a 2 cm thick steak was removed to measure pH and color, and another 2.5 cm thick steak was removed for the measurement of shear force and water-holding capacity. After harvesting, these steaks were vacuum-packed, stored at 4 °C, and collected on day 2 postmortem for analysis. Then, four 1.5 cm thick steaks were removed and vacuum-packed for Western blotting. For this purpose, samples were taken on days 1, 3, 5, and 7 postmortem, quickly frozen in liquid nitrogen, and placed at −80 °C until analysis. The pictorial description of steaks is shown in Figure 1.

### 2.2. Measurement of Meat pH

A portable pH meter (Testo 205, Lenzkirch, Germany) was used to measure the pH of the meat. Before measuring pH, the calibration of the pH meter was performed using buffers that had been stored at room temperature (20 °C) and had pH values of 4.00 and 7.00. The pH was assessed 48 h following the slaughtering.

### 2.3. Evaluation of Meat Color and Shear Force

The color of the LTL muscle steaks was measured at four random places using a chroma meter (CM-600-D spectrophotometer, Osaka, Japan) and averaged for statistical analysis, making sure to exclude connective tissue and fat flakes. The D 65 illuminant, 10° standard observer, and 8 mm aperture were employed. The color of the meat was captured over the oxygen-permeable film after the calibration of the chroma meter with a white plate.

Shear force was measured by placing the steaks in polypropylene bags and cooking them in a water bath until their temperature reached 72 °C. The core temperature was recorded using a temperature data logger (Multi Probe Data Logger, L93-2 L+, Hangzhou, China). After that, the packing was removed, and the steaks were cooled down to room temperature. To record the shear force values, cubes with dimensions of 1 × 1 × 4 cm were prepared and cut with a V-slot blade of the Texture Analyzer (Texture Analyzer, TA.XT plus^®^, Surrey, UK).

### 2.4. Measurement of Water-Holding Capacity

Water-holding capacity was computed using a method that was slightly altered from that of [22]. In short, 1.5 g of the sample was homogenized using 9 mL of the extraction buffer (pH 8.3, 10 mM DTT, 100 mM Tris) and then centrifuged (15,000× *g*) at 4 °C for 30 min. Then, the myofibrils underwent three rounds of washing in a pH 5.5 washing buffer containing 75 mM KCl, 2 mM EGTA (ethylene glycol tetraacetic acid), 100 mM MES (2-(N-morpholino) ethanesulfonic acid hydrate), and 2 mM MgCl_2_. The supernatant was disposed of after centrifuging the solution for ten minutes at 2400× *g*. The weight of myofibril water was measured by weighing the myofibril pellets before and after they were dried for a whole night at 100 °C in an oven. The salt residue from the washing buffer was taken into account when adjusting the weight of the dried myofibril protein pellets. The water in one gram of protein was used to calculate the relative water-holding capacity of myofibrils.

### 2.5. Determination of Glycogen Content

Glycogen was measured at 620 nm using a spectrophotometer (SpectraMaxR 190, Molecular Devices, San Jose, CA, USA) by following the protocols mentioned in the glycogen kit (NJBI, A043, Nanjing, China). In short, NaOH was added with 70 mg of the frozen sample (3:1). The tubes were then covered with a perforated membrane and placed into boiling water for 20 min. After cooling under tap water, distilled water (1.12 mL) was added to each tube. The distilled water was used as blank, and the concentration of standard solution was used as 0.01 mg/mL.

### 2.6. Determination of Lactate Content

The quantity of lactate was measured using a specific kit (NJBI, A019-2, Nanjing, China). About 1 g of frozen sample was transferred into the 9 mL of ice-cold 0.9% NaCl and then homogenized for 3 × 15 s with an intermediated rest of 30 s. Then, samples were centrifugated at 2500× *g*, 4 °C for 10 min. The protein concentration was measured using the supernatant (PCC BCA assay-kit, Rockford, IL, USA). The distilled water was used as blank, and the concentration of standard solution was used as 3 mmol/L. The lactate was measured at the wavelength of 530 nm.

### 2.7. Sarcoplasmic Protein Extraction

The proteins were isolated using the methodology outlined in earlier research [23], with a few minor adjustments. Briefly, 12 milliliters of chilled extraction buffer comprising protease inhibitor (Complete Protease Inhibitor Tablet, Germany; one tablet per 10 milliliters) was used to homogenize two grams of the frozen meat sample for 30 s with the help of a homogenizer (Ultra TurraxT10, IKA Labortechnik, Staufen, Germany). This process was repeated three times, with a 15 s break in between on ice. After that, the homogenate was centrifuged for 20 min at 16,000× *g*, and the temperature was set at 4 °C (Neofrosense 15R, Heal Force, Shanghai, China). Sarcoplasmic-protein-containing supernatant was poured into a new tube, whereas the myofibrillar-protein-containing pellet was discarded. Sarcoplasmic proteins were diluted 20 times to quantify protein content using the above-mentioned BCA assay kit. The spectrophotometer measurements were taken at 562 nm.

### 2.8. Sodium Dodecyl Sulfate Polyacrylamide Gel Electrophoresis (SDS-PAGE)

To prepare the final protein concentration of 1 µg/µL for SDS-PAGE, proteins were combined with an equivalent volume of the loading buffer (250 g/L glycerin, 1 g/L bromophenol blue, 40 g/L SDS powder, and 100 mmol/L Tris) and distilled water. Following shaking, the mixture was heated for five minutes in boiling water, cooled under running tap water, and centrifuged for two minutes, and the supernatant was taken and stored at −80 °C for examination. Subsequently, the polyacrylamide gels were loaded with 10 µg of the proteins. The gels were made up of 4% stacking gel, 8% separating gel (for HSP 20), or 12% separating gel (for HSP 70 and HSP 90). The SDS-PAGE marker of molecular weight (Thermo Fisher Scientific, Rockford, IL, USA) comprising phosphorylated and non-phosphorylated proteins was loaded onto the first well to point out the molecular weight. As a control, the samples taken from the normal beef after 24 h postmortem were put into the first well of each gel. At room temperature, gels were run with a tetra cell electrophoresis system (Mini-PROTEAN, BIO-RAD, Hercules, CA, USA) set to 70 V for the stacking gel and increased to 120 V for the separating gel.

### 2.9. Immunoblot Analysis

Following electrophoresis, proteins were transferred by a Mini Transblot Cell (BRL, Hercules, CA, USA) at 90 V for 90 min, using transfer buffer (20% methanol, 1.4% glycine, and 25 mM Tris–HCl, pH 8.3), from the gels to the 0.45 µm PVDF membrane (Merck Millipore, Billerica, MA, USA). At room temperature, 3% bovine serum albumin in TBS with 0.05% Tween 20 (TBST) was used to block the membranes for two hours after being cleaned three times in TBS (pH 7.5, 150 mM NaCl, 10 mM Tris). Following incubation with the monoclonal anti-HSP27 (Abcam, Cambridge, UK, 1:5000, ab2790), anti-HSP70 (Abcam, 1:1000, ab2787), and anti-HSP90 (Abcam, 1:1000, ab13492), the membranes were washed three times in TBST for the period of 20 min. Membranes were treated with goat anti-mouse IgG conjugated with horseradish peroxidase (ab6789, Abcam, 1:5000) overnight at 4 °C after blocking for 2 h. The ChemiDoc (BioRad, Hercules, CA, USA) was utilized to scan proteins on membranes using the substrate for clarity (1,705,061, Bio-Rad, Hercules, CA, USA). The quantity one software (4.62, Bio-Rad, Hercules, CA, USA) was utilized to examine the quantity of protein bands.

### 2.10. Statistical Analysis

The means ± standard deviations of the data are displayed. The SPSS (SPSS Statistics 21.0, Chicago, IL, USA) program was used to conduct statistical analysis. The ANOVA approach was used for data analysis, and Duncan’s multiple-range test was employed for the post hoc multiple comparisons for observed means. The level of significance was set at 5%.

## 3. Results

### 3.1. Meat Quality Measurements

The pH, shear force, water-holding capacity, and color coordinate (L*, a*, and b*) values of LT muscles of all three meat groups are shown in Table 1. Results presented notable differences in pH levels across the three groups (*p* < 0.05), and pH values were higher in the typical DFD beef than those of the values of the other two groups. Color L* was higher in the normal beef when compared with the other two groups. However, a* and b* were similar between the atypical DFD beef and the normal beef, but these values were lower in the case of the typical DFD beef (*p* < 0.05). Meat shear force values were higher in atypical DFD beef, and no difference was found between the values of the normal and typical DFD beef. Typical DFD beef showed higher values of water-holding capacity when compared with the atypical DFD and normal beef (*p* < 0.05).

### 3.2. Glycolytic Metabolite Measurements

Meat glycolytic metabolites that include glycogen and lactate contents of LT muscles are shown in Table 1. Anaerobic glycolysis converts the glycogen into lactate, and the accumulation of lactate causes a decline in muscle pH in postmortem muscles. In this way, meat pH value is linked with the glycogen and lactate contents. In the current study, it was found that typical DFD beef that had higher pH values showed lower glycogen and lactate contents; however, normal beef that had lower pH values presented higher glycogen and lactate contents in comparison with the other two groups (*p* < 0.05).

### 3.3. Expression of HSP27, HSP70, and HSP90

The expression of HSP27, HSP70, and HSP90 of LT muscles is shown in Figure 2 and Figure 3. Atypical DFD beef presented higher expression of HSP27 compared with the expressions of the normal and typical DFD beef on day 1 postmortem (*p* < 0.05, Figure 2). However, no difference was found between the typical DFD, atypical DFD, and normal beef samples on days 3, 5, and 7 postmortem. In atypical DFD beef, expression of HSP27 was higher on day 1 postmortem, declined from day 1 to 3 postmortem, and then remained unchanged from day 3 to 7 postmortem. In the typical DFD beef, the expression of HSP27 on day 1 was higher than that on days 5 and 7 (*p* < 0.05); however, the expression on day 3 was the same as the expressions on days 1, 5, and 7 of postmortem. The difference between normal beef among the storage times was not significantly different.

On day 1 postmortem, typical DFD beef showed higher expression of HSP70 as compared with the levels of the normal and atypical DFD beef (*p* < 0.05, Figure 3). On day 3 postmortem, the levels of HSP70 were similar between atypical DFD and normal beef but lower compared with the typical DFD beef. However, no difference was found between typical DFD, atypical DFD, and normal beef on days 5 and 7 postmortem. In the typical DFD, the expression of HSP70 on day 1 was higher than that on days 5 and 7 postmortem (*p* < 0.05); however, the expression on day 3 was the same as the expressions on days 1, 5, and 7 postmortem. In atypical DFD beef, the expression of HSP70 was higher on day 1 postmortem, declined from day 1 to 3 postmortem, and then remained unchanged from day 3 to 7 postmortem. In normal beef, expressions of HSP70 were not significantly different among the storage times.

Typical DFD beef showed a higher level of HSP90 than the level of the normal beef on day 1 postmortem (*p* < 0.05, Figure 4), and the level of the atypical DFD was comparable with the typical DFD and normal beef. Expressions of HSP90 were similar between atypical DFD and typical DFD beef on day 3 postmortem but higher than the expression of the normal beef (*p* < 0.05). However, no difference was found between typical DFD, atypical DFD, and normal beef on days 5 and 7 postmortem. In normal beef, the expression of HSP90 was decreased from day 1 to 3. In atypical DFD beef, the expression of HSP90 was higher on day 1 postmortem than that on days 5 and 7, and expression on day 3 was the same as on days 1, 5, and 7 postmortem. In typical DFD beef, the expression of HSP90 decreased from day 1 to 3 postmortem, remained stable from day 3 to 5 postmortem, and decreased from day 5 to 7 postmortem (*p* < 0.05).

### 3.4. Relationship of HSPs with Meat Quality and Glycolytic Metabolite

Pearson’s correlation was applied to find the relationship of HSP27, HSP70, and HSP90 with meat pH, shear force, water-holding capacity, color (L*, a* and b*), glycogen, and lactate contents (Table 2). The results showed that HSP27 presented just one significant correlation and that was a positive correlation with the shear force values (*p* = 0.006). HSP70 was directly correlated with pH (*p* = 0.02) and water-holding capacity (*p* = 0.018) and indirectly correlated with color a* (*p* = 0.042), b* (*p* = 0.05), glycogen (*p* = 0.008), and lactate (*p* = 0.018) contents. However, HSP90 showed a positive correlation with pH (*p* = 0.001) and water-holding capacity (*p* = 0.004) and a negative correlation with the color L* (*p* = 0.001), a* (*p* = 0.009), b* (*p* = 0.007), glycogen (*p* = 0.024), and lactate (*p* = 0.003) contents.

## 4. Discussion

The formation of DFD beef is a complicated process and is influenced by multiple variables, such as stress before slaughter and postmortem metabolism. Comprehending the molecular mechanisms behind the formation of DFD beef is crucial for managing this flaw, improving meat quality, and ensuring consumer satisfaction. In this work, we examined the function of HSPs in the advancement of DFD beef by testing the role of HSP27, HSP70, and HSP90 in the formation of ultimate meat quality characteristics. The findings of the investigation deepen our knowledge about the evolution of DFD beef and highlight the significance of HSPs in regulating meat quality.

Higher pH values in typical DFD than other beef groups are due to lower glycogen levels, which also show that there is not enough energy to create enough lactate and ultimate pH levels in typical DFD [16,24]. Regarding meat color, Liu et al. [25] and McKeith et al. [26] observed a desirable color of normal beef, which is in accordance with the results of the current investigation. Recent research [27] found that the main reasons for the decreased light scattering in the DFD beef were smaller sarcomeres, greater myofiber diameters, and larger lattice spacing. Previous research [28] revealed that the dark color of the DFD beef was primarily due to its higher WHC and increased levels of myoglobin and hemoglobin in the meat.

Normal and typical DFD beef have similar but lower shear force values than atypical DFD beef. This is because medium-pH meat has higher heat-shock protein levels, which shield the myofibrillar proteins from enzymatic degradation and ultimately reduce tenderness [15]. Similar findings were recorded in the previous literature that DFD meat with a high ultimate pH (>6.2) is usually of similar or greater tenderness compared to normal meat with an ultimate pH of about 5.5 [15,29]. However, in the medium-pH range of 5.8 to 6.2, less tender meat is generated [15,16]. Watanabe et al. [29] discovered that meat with a medium pH had fewer titin and nebulin breakdown products. On the other hand, Pulford et al. [8] suggested that in medium-pH meat (5.7 < pHu < 6.3), heat-shock proteins shield myofibrillar proteins from denaturation, and in doing so, the enzymatic cleavage by proteases is hindered, which ultimately helps in the production of tough meat. In the current investigation, atypical DFD beef exhibited higher shear force values and belonged to the medium-pH group. On the other hand, the main reason why normal beef has a lower water-holding capacity than typical DFD beef is because meat loses water-holding capacity when its pH approaches 5.5. After all, at this pH, myofibrillar proteins reach near their isoelectric points where positive and negative charges become equal [30]. It is worth mentioning that atypical DFD beef presented a higher shear force value, and WHC as compared with normal beef. Therefore, in the current study, tenderness was not related to the WHC of the meat, and the general perception about DFD meat that it becomes soft after cooking because it has more water contents was not applicable. The same results were also found by Holdstock et al. [15].

Typical DFD beef was found to have lower levels of lactate and glycogen, which suggested that the meat lacked the necessary muscular energy to reach an adequate ultimate pH [16]. These results were somewhat consistent with those of [15], who discovered that normal beef had equal glycogen reserves to atypical DFD beef but higher than typical DFD meat. The depletion of glycogen can occur as a result of pre-slaughter activities that cause physical exhaustion or set off the fight, flight, and fright response in animals [15,16]. These activities include long transportation, unfavorable weather conditions, feed restrictions, mixing animals from unfamiliar groups, and long lairage times [31]. Under these stress situations, catecholamines are more likely to be produced, which can drain glycogen by increasing the rate of glycolysis by activating the creatine phosphokinase, glycogen phosphorylase, and lactate dehydrogenase [32,33]. However, as lactate is the end-product of postmortem muscle glycolysis, our results showed that the amount of lactate and the concentration of glycogen showed a similar pattern, i.e., both were lower in typical DFD beef [34].

According to the current study, a higher level of HSP27 in atypical DFD beef is due to the molecular chaperone effect of sHSPs that protect the cell during stress conditions. Muscle cells in postmortem ischemia circumstances are subjected to apoptosis, a form of programmed cell death [35]. sHSPs are thought to be up-regulated in critical regions within the cell in response to impending cell death and have been found to play cytoprotective roles under adverse conditions [36]. Secondly, the up-regulation of HSP27 inhibits apoptosis [36] and consequently constitutes an obstacle to the meat maturation phase in atypical DFD beef.

On day 1 postmortem, higher levels of HSP70 and HSP90 in the typical DFD as compared with the normal beef were found. As described previously, HSP builds up in muscles with repetitive exercise or stress conditions [37]. Therefore, it was hypothesized that increased expressions of heat-shock proteins (HSP70 and HSP90) would play a part in the cell repair caused by pre-slaughter stress circumstances in DFD beef and demonstrated the role of pre-slaughter stress conditions in the formation of typical DFD beef.

The protective role of HSP27 in muscle cells that shields myofibrillar proteins from degradation in postmortem muscle, resulting in varying meat quality, may be the cause of the positive correlation between HSP27 and the shear force values [15].

The HSP70 and HSP90 presented a direct relationship with the pH, however an inverse relationship with glycogen and lactate contents. In other words, HSP70 and HSP90 showed a direct relationship with postmortem energy metabolism. Muscle glycogen content, which is influenced by several variables such as transport, weather and season, diet, and psychological stresses due to changes in the environment, affects the pH postmortem. Several stressors have been demonstrated to create abnormal pH and trigger the production of HSPs in skeletal muscle, including oxidative stress, diet restriction, heat stress, and bacterial infection [38,39]. But the reaction of animals to stress might differ greatly from one to another. The variability in how well animals adapt to changes in pH and their environment may be reflected in the expressions of HSPs. HSPs considered molecular chaperones and play a crucial role in maintaining the homeostasis of the cells and controlling apoptosis with their actions beginning right after the slaughtering of the animals [2]. It has been suggested that the higher production of HSP70 and HSP90 in typical DFD beef compared to normal beef results from the repair of denatured proteins and the restoration of cell membrane function, both of which improve water-holding capacity [3].

In the present study, HSP70 and HSP90 presented a positive relationship with the water-holding capacity. Numerous variables, such as the extent of extracellular space, the structure of muscle cells, and the net charge of proteins and cellular components (cytoskeletal linkage or myofibrils) might influence the retention of entrapped water in postmortem muscle [23]. It has been suggested that postmortem intracellular water loss is also related to changes in the integrity of the muscle membrane [40]. Under stressful circumstances, HSP70 has been shown to connect with membrane phospholipids and preserve the structural integrity and state of cell membranes [6]. By interacting with membrane phosphatidylcholine, HSP90 has been shown to perform a protective effect against phospholipase2 lipolysis [41]. The ideal pH range for the binding of HSP90 with the membrane is more than 6, as in DFD beef. On the other hand, in addition to having lower HSP90 levels, low-pH muscles (normal beef) may also have reduced HSP90 binding capacity to the membrane, which results in a reduction in water-holding capacity and increased drip loss [42,43]. Due to more drip loss, light reflects from the meat surface instead of absorbing, which increases the lightness of meat [44], with lower expression of HSP70 or HSP90. Moreover, HSP90 may block phospholipid inversion throughout the apoptotic process due to its anti-apoptotic function, delaying the rate at which pH declines.

The relationship between HSPs and different meat quality traits sheds light on how stress response impacts the quality of beef in normal, atypical, and typical DFD meat. The fact that HSP27 and shear force have a positive correlation and HSP70 and HSP90 with postmortem muscle metabolism is demonstrated by their direct correlation with pH and water-holding capacity, as well as their indirect effects on color, glycogen, and lactate contents. These relationships highlight how crucial HSPs are for controlling the quality of meat. A better comprehension of the behavior of the HSPs may present chances to improve beef quality control, possibly by reducing the prevalence of DFD and by enhancing meat qualities like color and tenderness.

Overall, our work sheds important light on the intricate interactions that occur between postmortem metabolism, heat-shock proteins (HSPs), and the evolution of meat quality in dark firm dry (DFD) beef. The observed variations in shear force values may be explained by the possible protective function of small HSPs in maintaining myofibrillar integrity, as shown by the higher levels of HSP27 in atypical DFD meat. On the other hand, the increased expression of HSP70 and HSP90 in normal DFD beef may be a sign of pre-slaughter stress and might be related to cellular repair processes. These results emphasize the role that HSPs play in preserving the qualities of meat quality and show how pre-slaughter circumstances affect postmortem metabolism. Furthermore, the positive associations shown between HSPs and the water-holding capacity of meat highlight how important these proteins are in controlling the integrity of muscle membranes and maintaining cellular homeostasis. The results of the study indicate that differences in the expression of HSPs may reflect the differences in the ability of an animal to adapt to the stress conditions, highlighting the need for more research to clarify the biochemical processes involved. Moreover, further exploring the molecular mechanisms by which HSPs affect ultimate meat quality could help in the development of novel approaches to improve meat quality and lower the prevalence of DFD in the beef industry.

The current study significantly advances our knowledge about the molecular processes that play a role in the production of DFD beef and emphasizes the significance of taking pre-slaughter stress factors into account when managing the quality of meat. Subsequent investigations concerning the regulation of HSP expression may present innovative methods for enhancing meat quality and guaranteeing sustainable beef production.

## 5. Conclusions

The levels of HSP27 were higher in atypical DFD beef, and it presented a direct association with the shear force as it protects the myofibrillar proteins from degradation in postmortem muscle. However, the levels of HSP70 and HSP90 were lower in normal beef, and both HSPs presented an inverse relationship with glycolytic metabolites as these heat-shock proteins adversely affect the postmortem metabolism. The direct relationship of HSP70 and HSP90 with the water-holding capacity may be due to the protective action of these proteins on the cell membrane that helps to retain water inside the cells. The less drip loss subsequently hinders light scattering and decreases the L* of meat with higher HSP70 and HSP90 levels. The current study was performed on LT muscles, and its results may not apply to other muscles with different myofiber compositions and metabolic properties; therefore, more investigation is required to clarify how DFD appears in different muscles throughout the carcass. Overall, these results may help to predict and manage variations in meat quality characteristics, especially under stress conditions. In the future, conducting a metagenomic analysis of the pathways could strengthen the findings of the current study.

## Figures and Tables

**Figure 1 foods-13-02965-f001:**
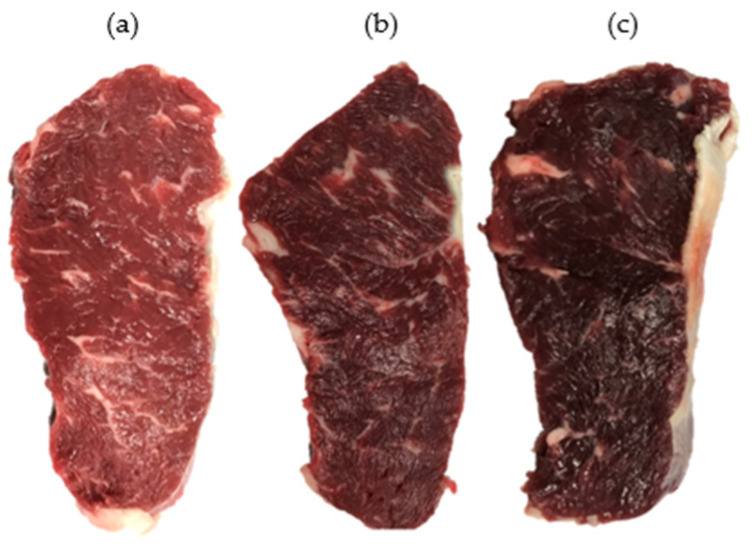
Pictorial description of normal (**a**), atypical DFD (**b**), and typical DFD (**c**) beef. The pictures were taken after the blooming of freshly cut samples at 4 °C for 45 min.

**Figure 2 foods-13-02965-f002:**
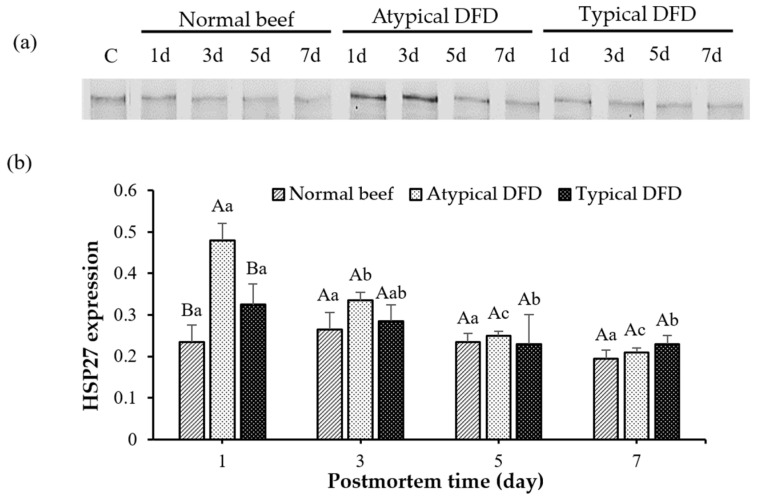
Western blotting analysis of HSP27 of normal beef, atypical DFD beef, and typical DFD beef on days 1, 3, 5, and 7 postmortem (**a**) and their relative intensities (**b**). C represents the control samples. A,B: different letters at specific postmortem times indicate statistical differences (*p* < 0.05) between groups. a–c: different letters within the same group indicate statistical differences (*p* < 0.05) between postmortem times. The data are presented as means ± standard deviations.

**Figure 3 foods-13-02965-f003:**
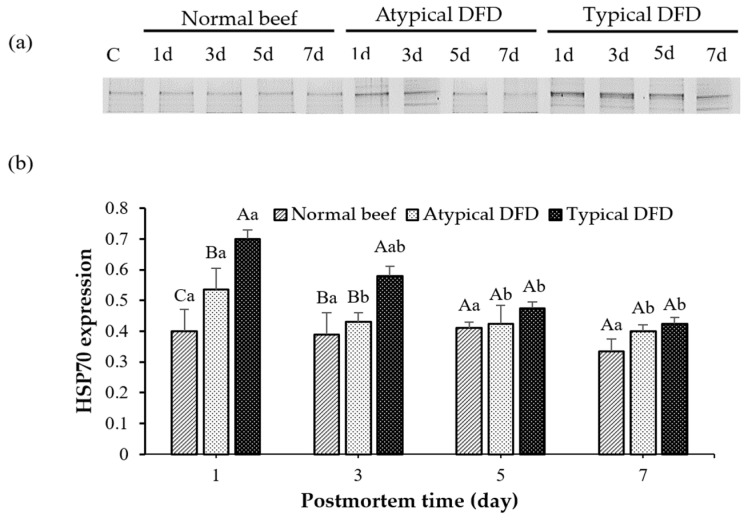
Western blotting analysis of HSP70 of normal beef, atypical DFD beef, and typical DFD beef on days 1, 3, 5, and 7 postmortem (**a**) and their relative intensities (**b**). C represents the control samples. A–C: different letters at specific postmortem times indicate statistical differences (*p* < 0.05) between groups. a,b: different letters within the same group indicate statistical differences (*p* < 0.05) between postmortem times. The data are presented as means ± standard deviations.

**Figure 4 foods-13-02965-f004:**
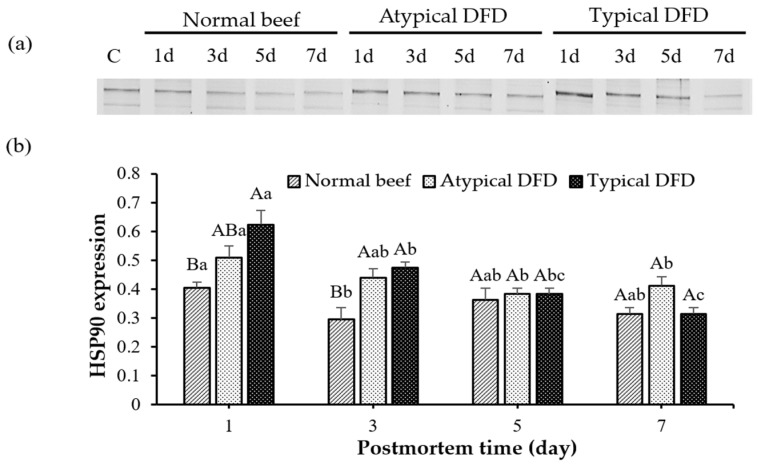
Western blotting analysis of HSP90 of normal beef, atypical DFD beef, and typical DFD beef on days 1, 3, 5, and 7 postmortem (**a**) and their relative intensities (**b**). C represents the control samples. A,B: different letters at specific postmortem times indicate statistical differences (*p* < 0.05) between groups. a–c: different letters within the same group indicate statistical differences (*p* < 0.05) between postmortem times. The data are presented as means ± standard deviations.

**Table 1 foods-13-02965-t001:** Meat quality characteristics, glycogen, and lactate contents of *longissimus thoracis* (LT) muscles of normal, atypical DFD, and typical DFD beef.

	Groups		
	Normal	Atypical DFD	Typical DFD
pH	5.47 ± 0.08 ^c^	5.82 ± 0.09 ^b^	6.21 ± 0.18 ^a^
L*	44.16 ± 2.16 ^a^	41.91 ± 1.20 ^b^	39.49 ± 2.31 ^c^
a*	14.68 ± 1.38 ^a^	13.01 ± 1.30 ^a^	10.66 ± 1.65 ^b^
b*	9.97 ± 2.37 ^a^	9.32 ± 1.49 ^a^	7.41 ± 1.22 ^b^
Shear force (N)	70.06 ± 3.11 ^a^	91.17 ± 4.66 ^b^	75.09 ± 7.15 ^a^
WHC (g water/g protein)	7.99 ± 0.74 ^c^	9.29 ± 0.94 ^b^	10.96 ± 1.34 ^a^
Glycogen (μmol/g)	8.11 ± 1.14 ^a^	5.28 ± 1.57 ^b^	2.89 ± 1.69 ^c^
Lactate (μmol/g)	197.56 ± 27.30 ^a^	174.23 ± 8.63 ^a^	141.41 ± 18.60 ^b^

^a–c^: within a row, different letters indicate statistical difference (*p* < 0.05) between groups. The data are presented as means ± standard deviations.

**Table 2 foods-13-02965-t002:** Pearson’s correlations between heat-shock proteins and meat quality attributes including glycogen and lactate contents of beef.

	pH	L*	a*	b*	Shear Force	WHC ^1^	Glycogen	Lactate
HSP27	0.259	−0.294	−0.145	0.017	0.935 *	0.272	−0.345	−0.183
	0.62	0.571	0.783	0.974	0.006	0.601	0.503	0.729
HSP70	0.883 *	−0.788	−0.828 *	−0.812 *	0.137	0.890 *	−0.925 *	−0.888 *
	0.02	0.063	0.042	0.05	0.796	0.018	0.008	0.018
HSP90	0.974 *	−0.995 *	−0.920 *	−0.932 *	0.159	0.945 *	−0.872 *	−0.952 *
	0.001	0.001	0.009	0.007	0.763	0.004	0.024	0.003

^1^ Water-holding capacity; significance at 0.05 is shown by the asterisks * on each association. The correlation coefficient (r) and *p*-value were used to represent the values.

## Data Availability

The original contributions presented in the study are included in the article, further inquiries can be directed to the corresponding author.

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
