# Peer review of "Role of Heat-Shock Proteins in the Determination of Postmortem Metabolism and Meat Quality Development of DFD Meat"

_foods, 2024, doi:10.3390/foods13182965_

Round 1
Reviewer 1 Report
Comments and Suggestions for Authors
The publication is interesting and original and brings new data to the meat science. However, I have a few comments:
1. The publication lacks a research hypothesis, which is most often placed at the end of the "Introduction" subsection.
2. The statement in lines 68-70 "The animals received the same treatment and nutrition before slaughtering. In accordance with the standard protocols of the slaughterhouse, all animals were slaughtered in the morning without the use of electrical stimulation or stunning." is too vague and general. Please provide more details.
3.The sentence on lines 73-74 is repeated on lines 84-85 "Calibration of the pH meter was done using buffers that had pH values ​​of 4.00 and 7.00 and were stored at room temperature (20 °C) 74 before the measurement." This is not necessary.
4.Table Header 1. There is no need to list all the characteristics in the table title. The table name should be more general, e.g. meat quality characteristics of different types of meat DFD.
5.Description above table 2. There is no need to list the values ​​of correlation coefficients, which are given in the table below. It would be better to mention the directions of dependencies and describe the connections between features in a general way, indicating trends or some directions of changes.
6. The sentence on line 286: "[23] discovered that meat with a medium pH had fewer titin and nebulin breakdown products." You cannot start the sentence with such a References number but with the author's surname and only then provide the References number.
7. The discussion in lines 283-291 does not explain the Shear force results. It should be better written. Why is the intermediate pH meat so hard? DFD meat is generally soft after heat treatment because it absorbs water. In Table 1 the results regarding WHC in typical DFD meat are consistent with this theory. DFD meat is hard but raw. What factors caused the intermediate pH meat to be harder. This should be better explained.
Author Response
Reference: foods-3200828
Title: “Role of heat shock proteins in the determination of postmortem metabolism and meat quality development of DFD beef” replaced with “Role of heat shock proteins in the determination of meat quality and postmortem metabolism of DFD meat”
Response to Reviewer 1 Comments
General Comment: The publication is interesting and original and brings new data to the meat science. However, I have a few comments:
Response: Thank you very much for taking the time to review our manuscript! Please find the detailed responses below and the corresponding revisions are highlighted with red-color text in the revised manuscript file. Your comments help a lot to improve our manuscript.
Comments 1: The publication lacks a research hypothesis, which is most often placed at the end of the "Introduction" subsection.
Response 1: Thank you for your comment! We have added the hypothesis of the study at the end of the introduction at lines 89-94.
Comments 2: The statement in lines 68-70 "The animals received the same treatment and nutrition before slaughtering. In accordance with the standard protocols of the slaughterhouse, all animals were slaughtered in the morning without the use of electrical stimulation or stunning." is too vague and general. Please provide more details.
Response 2: Thank you very much for your comment! We have revised the mentioned sentences to make them clear of meaning for the readers. The changes were made at lines 100-103.
Comments 3: The sentence on lines 73-74 is repeated on lines 84-85 "Calibration of the pH meter was done using buffers that had pH values ​​of 4.00 and 7.00 and were stored at room temperature (20 °C) 74 before the measurement." This is not necessary.
Response 3: Thank you very much for your comment! We have removed the repeated sentence from line 105.
Comments 4: Table Header 1. There is no need to list all the characteristics in the table title. The table name should be more general, e.g. meat quality characteristics of different types of meat DFD.
Response 4: Thank you for your suggestion. We have revised the header of the Table 1 based on your comment at line 236.
Comments 5: Description above table 2. There is no need to list the values ​​of correlation coefficients, which are given in the table below. It would be better to mention the directions of dependencies and describe the connections between features in a general way, indicating trends or some directions of changes.
Response 5: Thank you very much for your suggestions. Based on your comment, we have removed the values of correlation coefficient from the description and mentioned the relationship i.e., either its positive (direct) or negative (indirect) at lines 299-305.
Comments 6: The sentence on line 286: "[23] discovered that meat with a medium pH had fewer titin and nebulin breakdown products." You cannot start the sentence with such a References number but with the author's surname and only then provide the References number.
Response 6: Thank you very much for pointing out the correction. We have revised the corrected reference as “Watanabe et al. [29] discovered that….” at line 333. Additionally, we have corrected the mentioned mistake at several other places of the manuscript.
Comments 7: The discussion in lines 283-291 does not explain the Shear force results. It should be better written. Why is the intermediate pH meat so hard? DFD meat is generally soft after heat treatment because it absorbs water. In Table 1 the results regarding WHC in typical DFD meat are consistent with this theory. DFD meat is hard but raw. What factors caused the intermediate pH meat to be harder. This should be better explained.
Response 7: Thank you very much for your comment! In the previous literature, DFD meat with a high ultimate pH (>6.2) is usually of similar or greater tenderness compared to normal meat with an ultimate pH of about 5.5 (Bouton and others 1973a,b; Jeremiah and others 1991; Watanabe and others 1996; Grayson and others 2016). In the pH range of 5.8 to 6.2 less tender meat is generated (Dransfield 1981; Purchas, 1990; Jeremiah and others 1991; Watanabe and others 1996; Wulf and others 1996), which has been attributed to low titin and nebulin degradation rates (Watanabe and Devine 1996). These results have been recently confirmed in DFD meat from young cattle, where meat with intermediate pH was tougher than meat with >6.1 or normal meat (Holdstock and others 2014), and again more recently by Grayson and others (2016). Watanabe et al. (1996) discovered that meat with a medium pH had fewer titin and nebulin breakdown products. On the other hand, recently Pulford et al. (2009) suggested that in medium pH meat (5.7 < pHu < 6.3), heat-shock proteins shield myofibrillar proteins from denaturation and in doing so, the enzymatic cleavage by proteases is hindered, that ultimately helps in the production of tough meat. It has been mentioned in the manuscript as well at lines 330-337. All these results are aligned with the findings of the current study.
Furthermore, we have explained the general perception that DFD meat is soft after heat treatment/cooking because it absorbs water. We have revised the description in the discussion part of the manuscript to make it clear for the readers at lines 342-347.
- Bouton P, Carroll F, Fisher AL, Harris P, Shorthose W. 1973b. Effect of altering ultimate pH on bovine muscle tenderness. J Food Sci 38:816–20.
- Bouton PE, Fisher AL, Harris PV, Baxter R I. 1973a. A comparison of the effects of some post-slaughter treatments on the tenderness of beef. Int J Food Sci Tech 8:39–49.
- Dransfield E. 1981. Eating quality of beef. In: Hood D, Tarrant P, editors. The problem of dark-cutting in beef. Leiden, Boston: Martinus Nijhoff. P 344–58.
- Grayson AL, Shackelford SD, McKeith RO, King DA, Miller RK, Wheeler TL. 2016. Effect of degree of dark cutting on tenderness and flavor attributes of beef. J Anim Sci 94:2583–91.
- Holdstock J, Aalhus JL, Uttaro BA, Lopez-Campos O, Larsen IL, Bruce HL. 2014. The impact of ultimate pH on muscle characteristics and sensory attributes of the longissimus thoracis within the dark cutting (Canada B4) beef carcass grade. Meat Sci 98:842–9
- Jeremiah LE, Tong AKW, Gibson LL. 1991. The usefulness of muscle color and pH for segregating beef carcasses into tenderness groups. Meat Sci 30:97–114.
- Pulford, D.J.; Dobbie, P.; Vazquez, S.F.; Fraser-Smith, E.; Frost, D.A.; Morris, C.A. Variation in bull beef quality due to ultimate muscle pH is correlated to endopeptidase and small heat shock protein levels. Meat Sci. 2009, 83, 1-9.
- Watanabe A, Devine C. 1996. Effect of meat ultimate pH on rate of titin and nebulin degradation. Meat Sci 42:407–13.
- Wulf DM, Tatum J, Green R, Morgan J, Golden B, Smith G. 1996. Genetic influences on beef longissimus palatability in Charolais- and Limousin-sired steers and heifers. J Anim Sci 74:2394–405.
Reviewer 2 Report
Comments and Suggestions for Authors
Interesting and timely work in the context of meat quality.
I have some suggestions for the authors:
Improve the title to highlight the role of HSPs in defining meat quality.
L14: " the development of meat quality of DFD beef" is it only for DFD meat?
better clarify.
L 185- 186 : I suggest the authors to improve the commentary on the results: typical DFD meat has significantly lower a* and b* values.
Sez. 4: DISCUSSION
Specific comments
L274-282: Confusing comment, I invite you to explain better.
L 281: "DFD meat has a lower capacity to retain water". Are you sure? Statement contradicts your table.
L 297-307: I invite the authors to better contextualize the concepts expressed.
L 310-314: I find the contextualization of the bibliographic reference inappropriate. What should be highlighted instead is that HSPs are associated with an anti-apoptotic activity and consequently constitute an obstacle to the meat maturation phase.
L 316-322: Repetitive, concepts already expressed.
In general I find this section too long and the concepts are repeated several times. I invite you to review this section. There are many statements that only report experimental results. I would like to see something more added to your discussion that can be derived from your experimental results.
Author Response
Reference: foods-3200828
Title: “Role of heat shock proteins in the determination of postmortem metabolism and meat quality development of DFD beef” replaced with “Role of heat shock proteins in the determination of meat quality and postmortem metabolism of DFD meat”
Response to Reviewer 2 Comments
General Comment: Interesting and timely work in the context of meat quality. I have some suggestions for the authors:
Response: Thank you very much for taking the time to review our manuscript. We have considered all your comments and suggestions and carefully revised the manuscript accordingly. Please find the detailed responses below and revisions in the manuscript are highlighted with red-color text.
Comments 1: Improve the title to highlight the role of HSPs in defining meat quality.
Response 1: Thank you very much for your comment. We have revised the title of the study and now the title is “Role of heat shock proteins in the determination of the meat quality and postmortem metabolism of DFD meat” (lines 2-3).
Comments 2: L14: " the development of meat quality of DFD beef" is it only for DFD meat? better clarify.
Response 2: Thank you very much for your comment. We have revised the mentioned sentence as “…..the development of meat quality of normal, atypical DFD and typical DFD beef” in lines 14-15.
Comments 3: L 185- 186: I suggest the authors to improve the commentary on the results: typical DFD meat has significantly lower a* and b* values.
Response 3: Thank you very much for pointing out the mistake. We have corrected the mentioned mistake at line 232 and cross-checked the commentary on the results to avoid any such mistake.
Sez. 4: DISCUSSION
Specific comments
Comments 4: L274-282: Confusing comment, I invite you to explain better.
Response 4: Thank you very much for your comment. We have revised the whole paragraph to make it clear of meaning for the readers. The changes are made at lines 318-326.
Comments 5: L 281: "DFD meat has a lower capacity to retain water". Are you sure? Statement contradicts your table.
Response 5: Thank you for pointing out the mistake. We have cross-checked the reference and results of our study and corrected the mistake accordingly at line 325. So, “….dark color of the DFD beef was primarily due to its higher WHC….”.
Comments 6: L 297-307: I invite the authors to better contextualize the concepts expressed.
Response 6: Thank you very much for your comment. We have rephrased the mentioned paragraph to simplify the concepts and better understanding of the readers. The changes have been made at lines 352-361.
Comments 7: L 310-314: I find the contextualization of the bibliographic reference inappropriate. What should be highlighted instead is that HSPs are associated with an anti-apoptotic activity and consequently constitute an obstacle to the meat maturation phase.
Response 7: Thank you very much for your comment. We have rewritten the whole description based on your suggestion and have replaced the previous reference with an updated one at lines 365-369.
Comments 8: L 316-322: Repetitive, concepts already expressed.
Response 8: Thank you very much for your suggestion. We have removed the repetition and rephrased the paragraph at lines 371-372.
Comments 9: In general I find this section too long and the concepts are repeated several times. I invite you to review this section. There are many statements that only report experimental results. I would like to see something more added to your discussion that can be derived from your experimental results.
Response 9: Thank you for your suggestion. We have reduced the repetition and more effectively highlighted the key insights derived from our experimental results. At several places of the discussion, we have thoroughly revised the discussion to eliminate redundant statements and enhance the depth of analysis, focusing on the implications of our findings for meat quality characteristics in normal, atypical DFD, and typical DFD beef. The additions are highlighted with red-color text.
Reviewer 3 Report
Comments and Suggestions for Authors
This study investigated the relationship between three HSPs and DFD beef. Overall, the topic is very interesting, but the quality of the manuscript is fair and requires major revisions. These include adding more theoretical background in the introduction, providing more details about sample preparation and collection, and expanding the discussion. Incorporating additional experiments, such as proteomics and texture profile analysis (TPA), could significantly strengthen the results, increase the novelty, and enhance the knowledge gained from this study. Additionally, the English can be improved for clarity in some sentences. My comments are as follows:
General comment:
- Why was the longissimus thoracis selected as the sample? Is it due to its popularity for consumption or its risk for DFD? This reason should be stated in the introduction. Additionally, whether the results can be applied to other cuts of meat should be discussed in the conclusion for further application.
- Why were only three HSPs (26, 70, 90) selected for measurement in this study? I suggest that using advanced techniques, such as proteomics, could provide a comprehensive understanding of all HSPs and allow the authors to obtain a complete dataset.
- I suggest providing a picture of each sample to highlight the differences among the DFD samples and increase the reader’s engagement.
ABSTRACT
- Line13: “was planned” is not suitable. Please revise this word.
- When mention to the lower, higher, did it significant or not? I suggested to add P-value likes P<0.05, etc. at the end of this sentence.
INTRODUCTION
- There was no mention on typical and atypical DFD. How’s different among them. Please add this information.
- I suggest adding more theory related to post-mortem changes, such as pH, color, glycogen levels, etc. (particularly the parameters measured in this study), and how inappropriate changes can negatively affect meat quality (especially in terms of DFD). This would strengthen the introduction and provide a better foundation for the discussion.
MATERIAL AND METHODS
- Line76: Did those ref. referred to how to prepare sample as normal, typical and atypical DFD? If yes, I suggest to describe, particularly in brief, in the text since it was a core of this experiment.
- Why author use 5.70, 6.09 as the pH of criteria to separate among each group? More detailed on sample collection should provide likes how many sample per group, pH of individual, position to measure pH, etc.
- Did author measure pH at longissimus thoracis or any place, why?
- Line79: Stored for how long? Why?
- Line 79-81: About the sample collection for analysis was not clear. Which parameter should prepared as liquid nitrogen? Meat quality? How’s often to collect? Please rewrite them.
RESULTS AND DISCUSSION
- Why the results of shear force and WHC did not paralell. The highest shear force was found in atypical DFD, while the lowest WHC was found in normal meat? Please describe and add in discussion part. This discussion may highlight to the development of DFD in term of juiciness and tenderness of the meat.
- At this point, I suggest performing a texture profile analysis (TPA), as it is better than shear force. This is because TPA provides not only hardness but also measures parameters like springiness and juiciness of the sample.
- In the discussion section, please avoid using 'could be,' as the statements should be based on the results, not hypotheses.
- The results of the correlation analysis are interesting; however, I suggest the authors provide more discussion. What insights or advantages can be gained from understanding the correlation between each HSP and the measured parameters? How can this knowledge be applied further?
CONCLUSION
- Line394-395: The suggestion is not clear. How do HSPs provide a new perspective? As I mentioned earlier, conducting a metagenomic analysis of the pathways could strengthen the findings.
Author Response
Reference: foods-3200828
Title: “Role of heat shock proteins in the determination of postmortem metabolism and meat quality development of DFD beef” replaced with “Role of heat shock proteins in the determination of meat quality and postmortem metabolism of DFD meat”
Response to Reviewer 3 Comments
Comments 1: This study investigated the relationship between three HSPs and DFD beef. Overall, the topic is very interesting, but the quality of the manuscript is fair and requires major revisions. These include adding more theoretical background in the introduction, providing more details about sample preparation and collection, and expanding the discussion. Incorporating additional experiments, such as proteomics and texture profile analysis (TPA), could significantly strengthen the results, increase the novelty, and enhance the knowledge gained from this study. Additionally, the English can be improved for clarity in some sentences. My comments are as follows:
Response 1: Thank you for your constructive feedback on our manuscript! We really appreciate your time and suggestions for helping us to improve the manuscript. We have revised our manuscript according to your comments. We have provided more theoretical background in the introduction, provided more details about sample preparation and collection, and expanded the discussion. We also have highlighted our previous experiment on phosphoproteomics of sarcoplasmic and myofibrillar proteins, due to which the current study was designed to validate the role of heat shock proteins in the formation of meat quality. We have thoroughly revised our whole manuscript to improve its English grammar and sentence structure. We sincerely appreciate all of your comments and are confident that the revisions have enhanced the quality and readability of the manuscript.
General comment:
Comments 2: - Why was the longissimus thoracis selected as the sample? Is it due to its popularity for consumption or its risk for DFD? This reason should be stated in the introduction. Additionally, whether the results can be applied to other cuts of meat should be discussed in the conclusion for further application.
Response 2: Thank you very much for your suggestion. The longissimus muscle is the longest muscle in the animal body that runs parallel to both sides of the vertebrae. Furthermore, longissimus thoracis (LT) is the most commonly utilised reference muscle in meat quality tests. Secondly, it is a representative cut for consumer-relevant quality studies since it is one of the most popular and commercially significant beef cuts. The formation of DFD beef is mostly attributed to changes in stress, postmortem metabolism, and pre-slaughter handling, all of which have been shown to be especially susceptible to LT muscles in several investigations. Therefore, we selected LT muscle for its susceptibility to DFD and its relevance in assessing overall meat quality. The information has been added in the introduction part of the manuscript at lines 78-86.
Regarding the application of results to other meat cuts, as different muscles have different myofiber compositions and metabolic properties and susceptibility to stress. For example, glycolytic muscles like longissimus thoracis tend to develop DFD more frequently than oxidative muscles, such as the masseter. Therefore, these muscles may behave differently in postmortem studies. Some general trends may apply, but more investigation is required to clarify how DFD appears in different muscles throughout the carcass. The information has been added in the conclusion part at lines 452-455.
Comments 3: - Why were only three HSPs (26, 70, 90) selected for measurement in this study? I suggest that using advanced techniques, such as proteomics, could provide a comprehensive understanding of all HSPs and allow the authors to obtain a complete dataset.
Response 3: Thank you very much for your comment. In this study, we focused on HSP27, HSP70, and HSP90, as these specific proteins were selected based on findings from our previous experiments, which indicated their significant roles in stress response and meat quality in DFD beef (Ijaz et al., 2022). To ensure clarity, we have included this information in the introduction section to explain the rationale to readers (lines 50-53).
- Ijaz, M., Li, X., Zhang, D., Bai, Y., Hou, C., Hussain, Z., & Huang, C. (2022). Sarcoplasmic and myofibrillar phosphoproteins profile of beef M. longissimus thoracis with different pHu at different days postmortem. Journal of the Science of Food and Agriculture, 102(6), 2464-2471.
Comments 4: - I suggest providing a picture of each sample to highlight the differences among the DFD samples and increase the reader’s engagement.
Response 4: Thank you very much for your suggestion. We have provided the pictorial description of the samples in the manuscript at line 124-127 and described it in the text portion at lines 122-123.
ABSTRACT
Comments 5: - Line13: “was planned” is not suitable. Please revise this word.
Response 5: Thank you for your comment. We have revised the sentence based on the suggestion at line 13.
Comments 6: - When mention to the lower, higher, did it significant or not? I suggested to add P-value likes P<0.05, etc. at the end of this sentence.
Response 6: Thank you for your comment. We have added the P-value in the abstract portion at lines 19, 22, and 24.
INTRODUCTION
Comments 7: - There was no mention on typical and atypical DFD. How’s different among them. Please add this information.
Response 7: Thank you very much for your comment. We agree with the reviewer and have added the information about typical and atypical DFD beef in the introduction part at lines 58-62 and 67-70.
Comments 8: - I suggest adding more theory related to post-mortem changes, such as pH, color, glycogen levels, etc. (particularly the parameters measured in this study), and how inappropriate changes can negatively affect meat quality (especially in terms of DFD). This would strengthen the introduction and provide a better foundation for the discussion.
Response 8: Thank you very much for your suggestion. We have reviewed the previously available literature and added detailed information about the pH, colour, glycogen levels and tenderness of DFD meat samples. It helps us to strengthen the discussion part. The information has been added at lines 54-72.
MATERIAL AND METHODS
Comments 9: - Line76: Did those ref. referred to how to prepare sample as normal, typical and atypical DFD? If yes, I suggest to describe, particularly in brief, in the text since it was a core of this experiment.
Response 9: Thank you very much for your comment. Based on your suggestion, we have explained the process of sample preparation of normal, atypical and typical DFD according to the references. The changes have been made at lines 106-115.
Comments 10: - Why author use 5.70, 6.09 as the pH of criteria to separate among each group? More detailed on sample collection should provide likes how many samples per group, pH of individual, position to measure pH, etc.
Response 10: Thank you very much for your comment. These are the pH values at which samples behave differently regarding their glucose or glycogen levels, colour and tenderness. We have explained it in the introduction part (lines 54-70). Secondly, these values are also used by different studies to differentiate among normal, atypical and typical DFD beef samples, as mentioned in the materials and methods part (lines 106-110).
Based on the comment of the reviewer, we have added complete detail on the sample collection, including how many samples per group, the pH of the individual and the position to measure pH from lines 106-122.
Comments 11: - Did author measure pH at longissimus thoracis or any place, why?
Response 11: Thank you for your comment. The pH of LT muscles was measured between the 11th and 12th ribs, the information has been added at lines 105-106.
Comments 12: - Line79: Stored for how long? Why?
Response 12: Thank you for your comment. Based on your comment, complete detail about the sample transportation (lines 111-114) and storage (lines 115-122) has been mentioned in the manuscript for better understanding of the readers.
Comments 13: - Line 79-81: About the sample collection for analysis was not clear. Which parameter should prepared as liquid nitrogen? Meat quality? How’s often to collect? Please rewrite them.
Response 13: Thank you very much for your comment. From the posterior end of LT muscles, a 2 cm thick steak was removed to measure pH and colour, and another 2.5 cm thick steak was removed for measurement of shear force and water-holding capacity. After harvesting, these steaks were vacuum-packed, and stored in a refrigerator working at 4 °C and collected at day 2 postmortem for analysis. Then four 1.5 cm thick steaks were removed and vacuum-packed for Western blotting. For this purpose, steaks were removed from the refrigerator on the 1, 3, 5, and 7 postmortem days and samples were taken and quickly frozen in liquid nitrogen and placed at -80 °C for examination. We have added the complete information in the manuscript at lines 115-122.
RESULTS AND DISCUSSION
Comments 14: - Why the results of shear force and WHC did not paralell. The highest shear force was found in atypical DFD, while the lowest WHC was found in normal meat? Please describe and add in discussion part. This discussion may highlight to the development of DFD in term of juiciness and tenderness of the meat.
Response 14: It is worth mentioning that atypical DFD beef presented a higher shear force value, and WHC as compared with normal beef. Therefore, in the current study, tenderness was not related to the WHC of the meat and the general perception about the DFD meat that it becomes soft after cooking because it has more water contents was not applicable. The same results were also found by Holdstock et al. (2014). The information has been added at lines 342-347.
- Holdstock, J.; Aalhus, J.L.; Uttaro, B.A.; López-Campos, Ó.; Larsen, I.L.; Bruce, H. L. The impact of ultimate pH on muscle characteristics and sensory attributes of the longissimus thoracis within the dark cutting (Canada B4) beef carcass grade. Meat Sci. 2014, 98, 842-849.
Comments 15: - At this point, I suggest performing a texture profile analysis (TPA), as it is better than shear force. This is because TPA provides not only hardness but also measures parameters like springiness and juiciness of the sample.
Response 15: Thank you for your insightful suggestion regarding the use of texture profile analysis (TPA). We recognize the additional benefits TPA provides in evaluating parameters such as springiness and juiciness. The current study was specifically designed to verify the findings of our previous experiment (Ijaz et al., 2022), which focused on the expression of heat shock proteins about DFD beef. Shear force was chosen as it aligns with our previous work and provides reliable data on tenderness. We appreciate your feedback and will consider TPA for future studies.
- Ijaz, M., Li, X., Zhang, D., Bai, Y., Hou, C., Hussain, Z., ... & Huang, C. (2022). Sarcoplasmic and myofibrillar phosphoproteins profile of beef longissimus thoracis with different pHu at different days postmortem. Journal of the Science of Food and Agriculture, 102(6), 2464-2471.
Comments 16: - In the discussion section, please avoid using 'could be,' as the statements should be based on the results, not hypotheses.
Response 16: Thank you very much for your suggestion. We have removed the word “could be” at several places of the discussion to make the discussion more meaningful.
Comments 17: - The results of the correlation analysis are interesting; however, I suggest the authors provide more discussion. What insights or advantages can be gained from understanding the correlation between each HSP and the measured parameters? How can this knowledge be applied further?
Response 17: We appreciate your suggestion to go into more detail about the correlation analysis. Now we have added more details that will help to clarify the possible importance of these connections in determining that how HSP expression and meat quality characteristics relate to each other. The understanding of these results may help to predict and manage variations in meat quality characteristics, especially under stress conditions. The information has been added at lines 411-419.
CONCLUSION
Comments 18: - Line394-395: The suggestion is not clear. How do HSPs provide a new perspective? As I mentioned earlier, conducting a metagenomic analysis of the pathways could strengthen the findings.
Response 18: Thank you very much for your suggestion. We have updated the mentioned sentence in order to make it clear and relevant with the findings of the current study at lines 455-458.
Round 2
Reviewer 2 Report
Comments and Suggestions for Authors
Improved article and can be accepted.
Reviewer 3 Report
Comments and Suggestions for Authors
The author improved the manuscript as suggested and rebutted all queries with logical reasons.